# Joint Neuropsychological Assessment through Coma/Near Coma and Level of Cognitive Functioning Assessment Scales Reduces Negative Findings in Pediatric Disorders of Consciousness

**DOI:** 10.3390/brainsci10030162

**Published:** 2020-03-12

**Authors:** Erika Molteni, Katia Colombo, Valentina Pastore, Susanna Galbiati, Monica Recla, Federica Locatelli, Sara Galbiati, Claudia Fedeli, Sandra Strazzer

**Affiliations:** 1School of Biomedical Engineering & Imaging Sciences, and Centre for Medical Engineering, King’s College, London SE1 7EU, UK; 2Neuropsychological and Cognitive-behavioral Service, Neurophysiatric Department, Scientific Institute, I.R.C.C.S. Eugenio Medea, 23842 Bosisio Parini, Italy; valentina.pastore@lanostrafamiglia.it (V.P.); susanna.galbiati@lanostrafamiglia.it (S.G.); monica.recla@lanostrafamiglia.it (M.R.); claudia.fedeli@lanostrafamiglia.it (C.F.); 3Neurophysiatric Department, Scientific Institute, I.R.C.C.S. Eugenio Medea, 23842 Bosisio Parini, Italy; federica.locatelli@lanostrafamiglia.it (F.L.); sara.galbiati@lanostrafamiglia.it (S.G.); sandra.strazzer@lanostrafamiglia.it (S.S.)

**Keywords:** pediatric brain injury, vegetative state, unresponsive wakefulness syndrome, minimally conscious state, early intensive neuropsychological assessment, Coma/Near Coma Scale, Level of Cognitive Functioning Assessment Scale

## Abstract

The present study aimed to: (a) characterize the emergence to a conscious state (CS) in a sample of children and adolescents with severe brain injury during the post-acute rehabilitation and through two different neuropsychological assessment tools: the Rappaport Coma/Near Coma Scale (CNCS) and Level of Cognitive Functioning Assessment Scale (LOCFAS); (b) compare the evolution in patients with brain lesions due to traumatic and non-traumatic etiologies; and (c) describe the relationship between the emergence to a CS and some relevant clinical variables. In this observational prospective longitudinal study, 92 consecutive patients were recruited. Inclusion criteria were severe disorders of consciousness (DOC), Glasgow Coma Scale (GCS) score ≤8 at insult, age 0 to 18 years, and direct admission to inpatient rehabilitation from acute care. The main outcome measures were CNCS and LOCFAS, both administered three and six months after injury. The cohort globally shifted towards milder DOC over time, moving from overall ‘moderate/near coma’ at three months to ‘near/no coma’ at six months post-injury. The shift was captured by both CNCS and LOCFAS. CNCS differentiated levels of coma at best, while LOCFAS was superior in characterizing the emergence from coma. Agreement between scales was *fair*, and reduced negative findings at less than 10%. Patients with traumatic brain injury (TBI) vs. non-traumatic brain injury (NTBI) were older and had neurosurgical intervention more frequently. No relation between age and the level of consciousness was found overall. Concurrent administration of CNCS and LOCFAS reduced the rate of false negatives and better detected signs of arousal and awareness. This provides indication to administer both tools to increase measurement precision.

## 1. Introduction

The assessment of the state of consciousness (CS) in patients emerging from coma is a major challenge for all centers delivering acute and post-acute inpatient rehabilitation to victims of acquired brain injury (ABI). Due to the need for frequent inspection of the CS at the bedside and in optimal arousal state, behavioral assessment remains essential [1], despite a high rate of misdiagnoses [2]. A number of tools specifically designed for accurate behavioral assessment of the CS [3] have been validated [4,5,6,7,8], and some have become a standard [1,3]. In parallel, the availability of unprecedentedly refined diagnostic resources has fostered deeper exploration of the disorders of consciousness (DOC). This led to the reconsideration of the vegetative state (VS), now called unresponsive wakefulness syndrome (UWS) [9], to the definition of the minimally conscious state (MCS) [10,11], and to the distinction of MCS +/- depending on residual capabilities to follow commands and communicate [12]. To date, evidence of leftover command following is regarded as the key feature of good evolution of the CS, and is assigned a relevant prognostic value [12].

The pediatric field has only partially benefited from all these advancements. Novel assessment tools are generally designed for and calibrated on adult cohorts. Validation in children remains sporadic, sized on small samples and aimed to the ‘wide’ pediatric age [13,14]. A recent survey and the most updated guidelines have both highlighted a lack of appropriate tools and standards for pediatrics [1,15]. Additionally, the behavioral assessment is intrinsically harder, due to the yet limited behavioral repertoire of infants and very young children [16,17], the dependence of outcome on the age at injury [18,19], and the overlap of the evolution of the CS with the physiological growth curve [20,21,22]. Normative datasets for adjusting young children’s scores are often needed (e.g., [23]), though not always available, and the outcome prediction is then drawn by inference from adult data. However, at least among children with traumatic brain injury (TBI), there is initial support for the use of command following as a predictor for emergence to CS [24,25]. The Coma Recovery Scale—Revised [26] has emerged as the gold standard neurobehavioral measure for assessment of adult cases; nevertheless, it has only been adapted and tested on healthy children so far [27]. Albeit lacking wide validation in children, the Rappaport Coma/Near Coma Scale (CNCS) [28,29] has been successfully applied in a few pediatric cases and small cohorts [14,30,31]. Ease of administration, basic behavioral cognitive and motor requirements of items, and suitability for a wide spectrum of ages are the main reasons for its increasing application. These factors make CNCS a competing option to the Levels of Cognitive Functioning Assessment Scale (LOCFAS) [32]. Indeed, LOCFAS was previously validated in children with both traumatic and non-traumatic ABI by our group [33,34], but showed to be inapplicable to patients younger than 48 months in practice.

The aims of this study are:To describe the emergence to a CS in a sample of children and adolescents with severe brain lesions of traumatic and non-traumatic etiologies during the post-acute rehabilitation. The CNCS [28,29] and LOCFAS [32,35] were employed as tools for behavioral assessment;To explore the agreement between the CNCS and LOCFAS;To quantify the practical advantage of administering both CNCS and LOCFAS instead of only one tool.

## 2. Materials and Methods

### 2.1. Participants

Ninety-two patients with diagnosis of UWS/VS or MCS (according to [36], see Appendix A) were recruited from a cohort of children and adolescents with acquired brain injury (ABI) referred to the Scientific Institute Eugenio Medea for post-acute rehabilitation.

Inclusion criteria were: (i) age at first assessment between 0 and 18 years; (ii) time between injury and first assessment <3 months; (iii) documented evidence of a severe ABI with DOC of traumatic, anoxic, vascular or infectious etiology, as confirmed by a Glasgow Coma Scale (GCS) [37] score ≤ 8 at insult; (iv) absence of congenital pathology or disability previous to the injury; (v) medical records sufficiently detailed to determine the injury severity and neurological findings.

### 2.2. Measures

Patients received two neurobehavioral evaluations 3 and 6 months after injury (T0 and T1 respectively) ± 7 days tolerance. Patients younger than 48 months (*n* = 38) received assessment through the Rappaport Coma/Near Coma Scale (CNCS). Older patients received both the CNCS [28] and the Level of Cognitive Functioning Assessment Scale (LOCFAS) [32,35] (Figure 1). CNCS and LOCFAS were delivered in random order by two experienced neuropsychologists (K.C. and C.F.) blinded to each other and in the morning of the same day (when both were delivered). The recruiter (S.S.) remained blinded to all the assessments. Diagnosis, assessments, further bibliography and standardization of the rehabilitation treatment are detailed in the Appendix A. The relationship between the patients’ consciousness condition and CNCS and LOCFAS data was mapped (by E.M., Appendix A).

Separately, a theoretical framework linking the clinical status and CNCS and LOCFAS levels was established (F.L. and S.G.) on the basis of items/domains contained in each scale. The clinical status was labelled as either ‘coma’, ‘vegetative state (VS)’, ‘minimally conscious state minus (MCS-)’, ‘minimally conscious state plus (MCS+)’, or ‘emergence to a conscious state (CS)’. This theoretical framework was used to compare CNCS and LOCFAS on a common ground (Table 1).

The study was approved by the Ethics Committee of Istituto di Ricovero e Cura a Carattere Scientifico Eugenio Medea on 15 February 2008, and registered with code PS-RIABIL Ex56/05/14 and protocol 07/08-CE; informed consent was signed by all caregivers, in line with the Declaration of Helsinki and subsequent amendments.

Importantly, being associated to CNCS Level 4 and LOCFAS Level 1, ‘coma’ is clinically defined here as lack of eye opening [10] (and lack of any electroencephalographic modulation over 24 h), while at a neurobehavioral level, it is defined as the absence of any specific response to stimuli administered through the scales. Additionally, the distinction in between ‘MCS+’ and ‘MCS-’ was based on motor requests (command responsivity or command following) and not on verbal performance. Despite item 11 of the CNCS being successfully administered to all cases for probing intelligible verbalizations, verbal responsivity was systematically observed to be a less reliable marker than motor responsivity. This was especially seen when involving intelligible cues and in younger aged patients. This is because several patients gradually developed response codes different from verbalization, and mainly motor-based (eye blink, finger movements, etc.).

### 2.3. Data Analysis

Continuous data is presented as means, standard deviations (SDs) and percentages; discrete data is presented as medians and modes. Qualitative variables were investigated by chi-square statistics. Age was assumed to be normally distributed. Normality was assessed for average and total CNCS scores. Parametric tests were thus applied to these measures. Non-parametric tests were applied otherwise. Spearman’s rank correlation coefficients, adjusted for the age at injury, were obtained to determine the relationship between quantitative variables. Clinical agreement between LOCFAS and CNCS was measured through Cohen-Kappa score. Agreement was graphically displayed in matrices based on the mathematical concept of “confusion”. All statistical analysis was carried out using the SPSS statistical package v.25 (IBM Corp., Armonk, NY, USA). Significance was assumed at a *p*-value < 0.05; high significance at *p*-value < 0.001.

## 3. Results

### 3.1. Patients

A cohort of 92 consecutive patients (60 males, 65.2%) was recruited in the 2008–2016 period from children and adolescents who attended our institute for rehabilitation. Inclusion criteria were age between seven months and 18 years, and UWS/VS or MCS in the post-acute stage. No measure was missed as they were in-patients; hence, the database is complete. Two groups of patients were formed consisting of 44 patients with TBI and 48 patients with non-traumatic brain injury (NTBI). In this second clinical group, eight patients (16.7%) presented with brain lesions due to vascular etiology, 18 (37.5%) due infectious etiology, and 22 (45.8%) due anoxic etiology. All patients received CNCS administration at three and six months post-injury.

The clinical characteristics of the overall, TBI and NTBI groups are reported in Table 2.

In order to verify the homogeneity of the two subgroups (TBI vs. NTBI), statistical analysis was performed. Significant differences were found for the variables: “age at injury” (*t*-value = 6.8, *p* < 0.001) and “need for neurosurgery” (χ^2^ = 13.6, *p* < 0.001). Indeed, patients with TBI were older at the time of insult. They had also undergone a neurosurgical intervention more frequently than NTBI patients.

Fifty-four of the 92 patients received both CNCS and LOCFAS at T0 and T1. The clinical characteristics of this subgroup, including both TBI and NTBI patients, are reported in Appendix A.

### 3.2. The Level of Awareness and Responsivity at Three and Six Months after Injury

The level of awareness and responsivity of each patient was assessed three (T0) and six months after injury (T1). At T0, patients older than 48 months were scored through CNCS and LOCFAS. The majority of them were diversely allocated in Levels 1, 2 and 3 of the CNCS, which corresponds to ‘near, moderate and marked coma’. However, most of them scored at Level 2, or ‘generalized response’ (66.7% of the sample) of LOCFAS. At T1, almost half of the patients scored ‘no coma’ at the CNCS. At LOCFAS, half scored into the Level 3, or ‘localized response’, a quarter fell into Level 5 ‘inappropriate, not agitated behavior’, and one out of five remained at Level 2 ‘generalized response’. The probability to fall into a certain CNCS or LOCFAS level was not significantly influenced by the variable etiology. Patients younger than 48 months, as well as the overall cohort, showed distributions comparable to those observed in older children (see Appendix A). Figure 2 shows the confusion matrices (joint representation of scores at CNCS and LOCFAS) at T0 and T1, for patients aged older than 48 months. At T0, patients in CNCS Levels 2 and 3 ‘moderate and marked coma’ were mainly scored at LOCFAS Level 2 ‘generalized response’. Patients in CNCS Level 1 ‘near coma’ were split between LOCFAS Levels 2 and 3 ‘generalized and localized response’. At T1, patients in CNCS Level 0 scored between 3 and 5 at LOCFAS (‘localized response’ and above).

### 3.3. The Evolution of the State of Consciousness from Three to Six Months after Injury

In the total sample, the average CNCS Level was 2.9 (SD = 0.9) at T0. It decreased to 1.8 (SD = 0.8) at T1 (*t* = 11.0, *p* < 0.001), indicating that patients showed higher levels of awareness and responsivity at six months with respect to three months after injury. All 11 items of the CNCS significantly changed between T0 and T1 in the direction of higher awareness and responsivity. Nevertheless, medians remained unchanged for the olfactory item, and pain was already saturated at T0. For details see Appendix A. The LOCFAS provided analogous results (see Appendix A).

Figure 3 shows the comparison of the CNCS and LOCFAS levels at T0 and T1. According to CNCS evaluation, patients in Level 3 ‘marked coma’ at T0 mainly moved to Levels 2 and 1 ‘moderate and near coma’ at T1. Patients who were in Levels 2 and 1 ‘moderate and near coma’ at T0, chiefly moved to Levels 1 and 0 ‘near and no coma’ at T1. Considering LOCFAS, patients scoring at Level 2 ‘generalized response’ at T0 split between Levels 2 and 3 ‘generalized and localized response’ at T1. Patients scoring at Level 3 ‘localized response’ at T0 split between Levels 3 and 5 ‘localized response and inappropriate behavior’ at T1.

### 3.4. The Evolution of the State of Consciousness in Patients with Stable Score at Three and Six Months after Injury

Of the 92 patients in the cohort, 24 (26.1%) scored the same CNCS Level at T0 and T1. This indicated no change in their classification at the two time points. Nevertheless, in this subgroup, the average CNCS score (mean at T0 = 1.8; mean at T1 = 1.5; *t*-value = 3.6; *p* = 0.002) and total CNCS score (mean at T0 = 20.1; mean at T1 = 16.2; *t*-value = 5.3; *p* < 0.001) were significantly lower at T1 vs. T0. Of the 54 patients assessed with both scales, 18 (33.3%) failed to show a change in LOCFAS level at T1. Five patients showed both the same CNCS and LOCFAS score at T0 and T1 (see details in Appendix A).

### 3.5. Clinical Agreement between CNCS and LOCFAS

Out of 54 cases, clinical agreement between CNCS and LOCFAS at T0 was found for 34 (63.0%) patients (Cohen-K = 0.40). Ten cases were rated ‘MCS-’ according to the CNCS and ‘vegetative state’ with LOCFAS, among others. Of note, three cases were rated ‘MCS-’ according to the CNCS and ‘emerged from coma’ with LOCFAS. These patients received high scores for the *attention for environment* and *responses to stimuli* items in the LOCFAS scale. Clinical agreement at T1 was found for 27 (50.0%) patients (Cohen-K = 0.33). Ten cases were rated ‘emerged from coma’ according to CNCS and ‘MCS+’ with LOCFAS. Seven cases were rated ‘MCS-’ according to the CNCS and ‘vegetative state’ with LOCFAS. Seven cases were rated ‘MCS-’ according to the CNCS and ‘MCS+’ with LOCFAS, among others. For a full description, see Figure 4.

### 3.6. The Evolution of the State of Consciousness with Respect to the Initial Ability to Follow Commands

‘High command responsivity’ at T0 resulted in lower total CNCS scores at T0 (*t* = 7.8, *p* < 0.001) and at T1 (*t* = 2.9, *p* = 0.005) with respect to the ‘low command responsivity’ group. This indicated better CS at both T0 and T1 when ‘high command responsivity’ was initially observed (see Appendix A for more details).

### 3.7. The Evolution of the State of Consciousness in the Sample Divided by Etiology

Table 3 reports the scores of each CNCS item at the two assessment times (T0 and T1) and their comparison, both for patients with TBI and NTBI. The TBI group had 10 out of 11 items significantly shifted towards higher awareness and responsivity at T1 vs. T0. Only the two items probing *pain* had unchanged medians and modes, as the descriptors were already saturated at T0. Similarly, the NTBI group had all items significantly improved. Further details and scores for LOCFAS in TBI and NTBI are reported in Appendix A and related Appendix A.

### 3.8. Effect of Age

After correction by gender, the effect of age was tested through correlations between “age at injury” and CNCS level, total CNCS score and LOCFAS level at both T0 and T1. Analysis showed no relation between age and the level of consciousness.

### 3.9. Correlations between the Clinical Characteristics and the CNCS and LOCFAS Scales

Negative correlations were found between total CNCS score at T0 and Glasgow Outcome Score -Extended (GOS-E), neurosurgery, and feeding disorders. CNCS positively correlated with tracheotomy and paroxysmal sympathetic hyperactivity. LOCFAS level at T0 positively correlated with feeding disorders and GOS-E.

At T1, CNCS negatively correlated with GOS-E. LOCFAS level negatively correlated with paroxysmal sympathetic hyperactivity, and positively correlated with feeding disorders and GOS-E (see Appendix A).

## 4. Discussion

The evolution of the state of consciousness (CS) in 92 children and adolescents with a severe ABI due to traumatic and non-traumatic events was described through two different scales: CNCS and LOCFAS. The emergence from a DOC to CS was fathomed against several clinical variables at three and six months after injury. The aim was to understand the factors mostly influencing the evolution.

CNCS and LOCFAS were administered with no dropout, as none of the patients missed any sessions, nor died by the end of the study. No specific scale adaptation was needed. CNCS was applicable to the youngest patients, in agreement with previous cases of effective employment in pediatrics [14,30,31]. LOCFAS was administered to all 54 children older than 48 months.

The cohort progressed from overall ‘moderate/near coma’ at three months to ‘near/no coma’ at six months post-injury. Progression towards milder DOC corresponded to one net CNCS level decrease (i.e., improvement) on average. The LOCFAS provided similar results. This is in agreement with our previous work [33], independent studies [13,38,39], and systematic reports [40].

At three months, a large subgroup of patients who showed ‘generalized response’ in the LOCFAS could be differentiated into either ‘near’, ‘moderate’ or ‘marked coma’ through the CNCS. This hints towards the superior capability of the CNCS to further differentiate generalized responses into finer coma levels. Conversely, the LOCFAS offered better resolution at T1 for discriminating localized, confused/agitated and non-agitated responses. These were all grouped as ‘no coma’ by the CNCS.

In line with previous findings by our group [33], the probability to fall into certain CNCS or LOCFAS levels was comparable for TBI and NTBI survivors at both time points. One independent study reports trends close to equality [13].

The only CNCS item already saturated at three months after injury was pain. Our interpretation is that pain items are crucial to add sensitivity and provide discrimination over the highest (i.e., severest) levels of the scale. Conversely, they hardly contribute to the discrimination of the lowest (i.e., mild) levels. As a result, pain items are often observed to be the earliest items to reach saturation when the patient recovers throughout the spectrum of DOC from coma towards consciousness. Nociception can travel through multiple ascending tracts, and can occur in the absence of awareness of pain [41]. It is possible that our patients experienced non-specific pain at that time. Moreover, in patients with extensive thalamic damage, the activation of nociceptors could have triggered mainly reflexive motor withdrawal. Reflexive withdrawal can be behaviorally congruent per se, but its association to painful experience is often uncertain in this context [42]. This has important implications for analgesic treatment [43], as emphasized in recent recommendations [1].

The LOCFAS showed that the domains changing the least were ‘ability to perform self-care activities’, ‘time orientation’ and ‘ability to learn new information’. These abilities correspond to complete recovery of consciousness, and are commonly the latest to be regained. Considering the overall extreme severity of this cohort, a longer follow-up could have shown stronger improvements in these domains. Longer observations are needed to confirm this hypothesis.

One quarter of patients demonstrated no change in their CNCS level between the two measurements at three and six months, but they had average CNCS score improvement in the same period. The CNCS has essential duality: (1) CNCS level allows classification, i.e., assignment of one patient’s status to a category of consciousness, and (2) average and total CNCS scores allow evaluation of subtle clinical changes. Indeed, total CNCS score sounds out both the frequency and extent of behavioral signs in great detail. Changes in score of single CNCS items provides magnification on specific modifications of the (re)emerging behavioral repertoire. By fully exploiting information from these two measures, therapists can utilize the sensitivity and specificity of the CNCS to interpret changes in single items as a proof of efficacy of specific treatments.

The concurrent employment of CNCS and LOCFAS assessments reduced the number of patients showing no level modification at either scale from 25% to less than 10%. In our opinion, this result provides strong indication for the administration of both tools to pediatric patients in clinical practice. Joint administration of the CNCS and LOCFAS will increase measurement precision and possibly reduce false negatives.

Although the correspondence between scale scores and the patient’s clinical condition was not exact, CNCS and LOCFAS showed *fair* in-between agreement at both three and six months. At T0, three cases rated ‘MCS-’ by CNCS were rated ‘emerged from coma’ by LOCFAS. LOCFAS might have overestimated *attention for environment* and *responses to stimuli*. Alternatively, it might have captured sings of arousal and awareness which remained undetected by CNCS.

Children with the initial ability to follow commands later demonstrated overall better CNCS and LOCFAS scores, and thus CS, at six months. This confirms the adequacy and suitability of the MCS+/- nosology for pediatrics [44]. It also underpins the prognostic role of the early ability to follow commands, in line with findings in adults [12].

Conversely, the need for a tracheotomy and the occurrence of paroxysmal sympathetic hyperactivity appear to be the main factors of co-morbidity. While the penalizing effect of tracheotomy is well-established [45], prognostic evidence of paroxysmal sympathetic hyperactivity is still controversial [46,47]. However, a greater incidence among younger patients was found [48]. The presence and type of feeding disorder also correlated with CNCS and LOCFAS scores. Oral feeding implies a full and complex oral phase, which as a whole is considered a sign of consciousness. However, it is not known which of its components are necessary to infer the presence of ‘conscious’ swallowing as opposed to reflex [49].

In line with previous epidemiological literature [50], this study found that patients with TBI were older at the time of insult, and needed neurosurgical intervention more frequently than NTBI patients. No relation between age and the level of consciousness was found overall.

### Study Limitations

This study has some limitations. The time span under neurobehavioral investigation is three months (T1–T0); this period is clearly too short for long-term changes in the CS to be surveyed. The cohort size allowed us to study patients with TBI vs. NTBI; however, further subdivision of the NTBI group into specific etiologies was prevented by a loss in statistical power. Although previously applied in pediatrics, the LOCFAS and CNCS have not been adequately validated for such usage yet.

## 5. Conclusions

In children, CNCS and LOCFAS show *fair* agreement, and their concurrent employment reduces negative findings from 25% to less than 10%. This advocates for the joint administration of both tools. It will reduce false negatives and increase standardization and precision.

The CNCS proved to differentiate levels of coma at best. Particularly, it allowed discrimination between ‘extreme’, ‘marked’ and ‘moderate’ coma states. Such fine discrimination among severe patients would have remained undetected through the LOCFAS. Consequently, the CNCS should be employed with patients of extreme severity, or with those in the early stages of the recovery. The LOCFAS was superior in characterizing the emergence from coma, and particularly in disentangling confused/agitated and not agitated behavioral profiles. For this reason, maximal utility is envisaged in less severe cases, and in patients already well under way to recovery.

However, more validation is needed for measures that are appropriate for administration to children with DOC, and specific guidelines for assessing severe DOC are needed in pediatrics.

## Figures and Tables

**Figure 1 brainsci-10-00162-f001:**
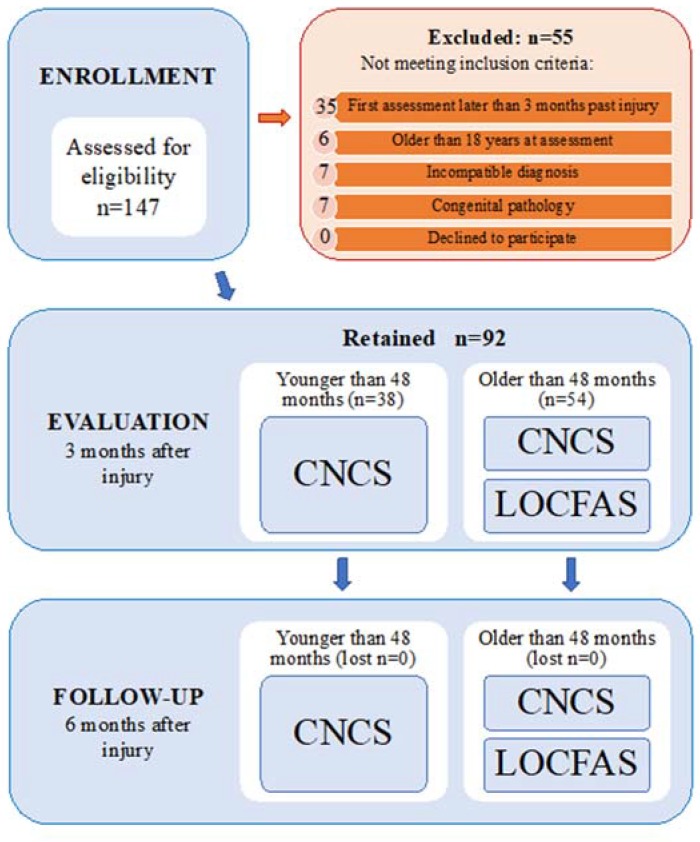
Flow diagram of the study indicating enrollment, exclusions, and group allocation. CNCS: Rappaport Coma/Near Coma Scale; LOCFAS: Level of Cognitive Functioning Assessment Scale.

**Figure 2 brainsci-10-00162-f002:**
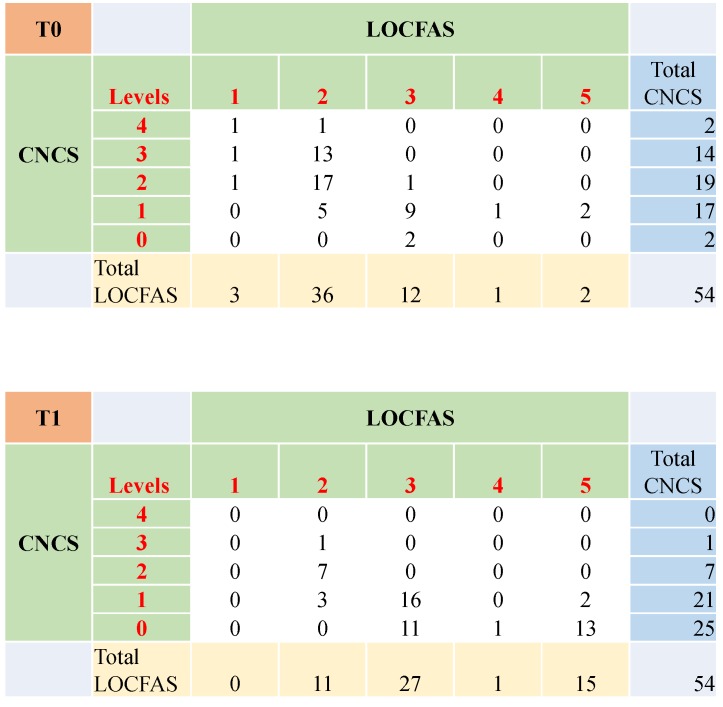
Confusion matrices at three months (T0) and at six months (T1) after injury for patients aged older than 48 months. At T0, patients in CNCS Levels 2 and 3 ‘moderate and marked coma’ were mainly scored at LOCFAS Level 2 ‘generalized response’. Patients in CNCS Level 1 ‘near coma’ were split between LOCFAS Levels 2 and 3 ‘generalized and localized response’. At T1, patients in CNCS 0 scored between 3 and 5 in the LOCFAS (‘localized response’ and above). Total LOCFAS is total number of patients who received a LOCFAS score; Total CNCS is total number of patients who received a CNCS score.

**Figure 3 brainsci-10-00162-f003:**
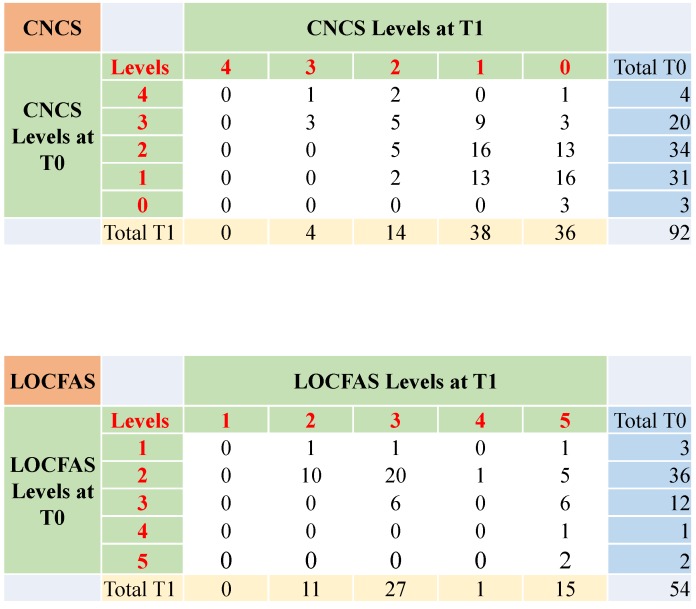
Evolution of the CS from T0 to T1, according to both CNCS and LOCFAS. Results refer to patients older than 48 months. The CNCS table shows that patients who were at Levels 2 and 1 ‘moderate and near coma’ at T0 chiefly moved to Levels 1 and 0 ‘near and no coma’ at T1. The LOCFAS table illustrates that patients scoring at Level 2 ‘generalized response’ at T0 split between Levels 2 and 3 ‘generalized and localized response’ at T1. Patients scoring at Level 3 ‘localized response’ at T0 split between Levels 3 and 5 ‘localized response and inappropriate behavior’ at T1.

**Figure 4 brainsci-10-00162-f004:**
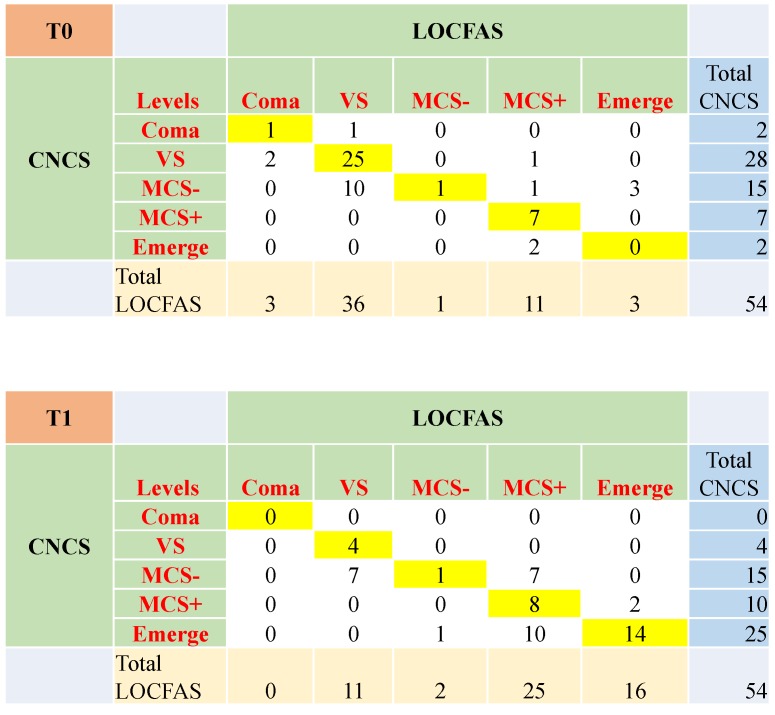
Clinical agreement between CNCS and LOCFAS at T0 and T1 for patients aged older than 48 months. At T0, agreement was 63.0% (yellow boxes, Cohen-Kappa = 0.40), and 50% at T1 (Cohen-Kappa = 0.33). VS: vegetative state.

**Table 1 brainsci-10-00162-t001:** Framework for theoretical association between clinical status and CNCS and LOCFAS levels. MCS: minimally conscious state.

Clinical Status	CNCS Level	CNCS Additional Criteria	LOCFAS Level	LOCFAS Additional Criteria
**Emerge**	Level 0	-	Level 5	
			Level 4	
**MCS+**	Level 1	And ‘Command responsivity’ is 0	Level 3	If specific score at item ‘Execution of commands’ is 2, 3 or 4
**MCS-**	Level 1	And ‘Command responsivity’ is 2 or 4	Level 3	If specific score at item ‘Execution of commands’ is 1
	Level 2	If ‘Auditory’ is 0, or if ‘Visual (item *n*.3 or 4)’ is 0, or both		
**Vegetative State**	Level 2	If ‘Auditory’ is not 0 and ‘Visual (item *n*.3 or 4)’ is not 0	Level 2	
	Level 3			
**Coma**	Level 4		Level 1	

CNCS: Rappaport Coma/Near Coma Scale; LOCFAS: Level of Cognitive Functioning Assessment Scale. MCS: minimally conscious state.

**Table 2 brainsci-10-00162-t002:** Clinical and demographic characteristics of the total sample and the two groups divided by etiology. SD: standard deviation; GCS: Glasgow Coma Scale; GOS-E: Glasgow Outcome Scale Extended.

	Total Sample (*n* = 92)		Traumatic (TBI) (*n* = 44)		Non-Traumatic (NTBI) (*n* = 48)		Statistics
	Mean	SD	Mean	SD	Mean	SD	*p* (*t*-Test)
**Age at injury (months)**	87.0	66.1	127.7	65.0	50.1	41.2	<1 × 10^−4^ **
	Median	Range	Median	Range	Median	Range	*p* (Wilcoxon test)
**GCS score**	4	3–8	4	3–7	4	3–8	0.892
**GOS-E score**	2	2–3	2	2–3	2	2–3	0.141
	*n*	%	*n*	%	*n*	%	*p* (2)
**Gender**							
**Male**	60	65.2	29	65.9	31	64.6	0.535
**Female**	32	34.8	15	34.1	17	35.4	
**Need for neurosurgery**	48	52.2	32	72.7	17	35.4	<1 × 10^−4^ **
**Tracheotomy**	46	50.0	27	61.4	20	41.7	0.060
**Feeding disorders**							
**Absence of disorders**	1	1.1	0	0.0	1	2.1	0.251
**Dysphagia**	5	5.4	4	9.1	2	4.2	
**Nasogastric tube (NGT)**	38	41.3	14	31.8	24	50.0	
**Percutaneous endoscopic gastrostomy (PEG)**	48	52.2	26	59.1	21	43.7	
**Paroxysmal sympathetic hyperactivity episodes**	38	41.3	20	45.5	18	37.5	0.529
**Motor impairment**							
**Absence of impairment**	1	1.1	1	2.3	0	0.0	0.370
**Motor retardation**	1	1.1	1	2.3	0	0.0	
**Quadriparesis**	89	96.7	42	95.4	47	97.9	
**Ataxia**	1	1.1	0	0.0	1	2.1	
**Previous rehabilitation**	4	4.3	2	4.5	2	4.2	0.572

** highly significant at *p*-value < 0.001.

**Table 3 brainsci-10-00162-t003:** Scores for 11 single CNCS items at T0 and T1, and results of the comparison of each item at T0 vs. T1. Results refer to the samples with TBI (*n* = 44) and NTBI (*n* = 48).

		Traumatic (TBI)					Non-Traumatic (NTBI)				
Item	Individual Parameters	T0		T1		Z (*p*)	T0		T1		Z (*p*)
		Median	Mode	Median	Mode		Median	Mode	Median	Mode	
**1**	**Auditory**	2	2	0	0	3.5 (0.001) *	2	2	0	0	4.2 (<0.001) **
**2**	**Command responsivity**	4	4	0	0	4.3 (<0.001) **	4	4	2	0	4.5 (<0.001) **
**3**	**Visual**	2	4	0	0	4.4 (<0.001) **	4	4	2	0	4.4 (<0.001) **
**4**	**Visual**	2	4	0	0	4.2 (<0.001) **	4	4	2	0	4.0 (<0.001) **
**5**	**Threat**	2	4	0	0	3.5 (<0.001) **	4	4	0	0	4.3 (<0.001) **
**6**	**Olfactory**	4	4	2	0	4.0 (<0.001) **	2	4	2	0	2.1 (0.033) *
**7**	**Tactile**	4	4	2	2	4.8 (<0.001) **	4	4	2	2	4.7 (<0.001) **
**8**	**Tactile**	2	0	0	0	4.2 (<0.001) **	0	0	0	0	2.1 (0.038) *
**9**	**Pain**	0	0	0	0	1.5 (0.132)	0	0	0	0	3.1 (0.002) *
**10**	**Pain**	0	0	0	0	2.3 (0.021) *	0	0	0	0	3.3 (0.001) *
**11**	**Vocalization**	4	4	2	2	4.1 (<0.001) **	3	4	2	2	4.3 (<0.001) **

** highly significant at *p*-value < 0.001. * significant at *p*-value < 0.05.

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
