# Peer review of "Joint Neuropsychological Assessment through Coma/Near Coma and Level of Cognitive Functioning Assessment Scales Reduces Negative Findings in Pediatric Disorders of Consciousness"

_brainsci, 2020, doi:10.3390/brainsci10030162_

Round 1
Reviewer 1 Report
Overall the paper by Molteni et al. presents interesting data and evaluation of two scales for determining level of consciousness in children. The study results are interesting and should prove useful for bedside use and monitoring, and the findings are important as accurate scales in critically injured children are needed.
However, the manuscript has some significant limitations that affect its acceptability as presented. These are listed below.
- Readability-1. The manuscript would benefit significantly from editorial review for semantic and syntactic use of words and terms that detract from what is being said in the paper. An example is on line 47 where the adjective "primal" is used in a presumed attempt to express the importance of behavioral assessment of consciousness. Although primal could refer to something fundamental in a different context, in my opinion it is not appropriately used in this context. Another example is the wording for Figure one – would it not be better that the table be called a “Consciousness” matrix versus a “confusional” matrix. From the perspective of the terms consciousness and confusional these are very different CNS states. And, the purpose of the paper is in evaluating degrees of consciousness.
- There are many other examples throughout the manuscript.
- Readability-2. The manuscript is cumbersome at times in how the various scales and results are discussed. The whole point of the study is to make the results accessible and useful to bedside clinician so an attempt to clarify wording and flow would be beneficial.
- Some more granular items for consideration.
- A definition of MCS and VS would be useful in the manuscript for the purpose of the comparisons in the paper. These terms are somewhat subjective and have different applied meaning – dependent on context and population.
- A patient flow diagram from recruitment, exclusion, and through to group allocation would be useful and easier to follow.
- When the two tests were administered the same day – was the order randomized or the same. Is there a potential negative or positive effect of test order? This is important since the tests are long and dependent on interpretation of response which may change in the face of fatigue or other variables.
- Similarly, whether the tests administered at the same time of day for all patients?
- The tables in the paper and supplemental section would benefit from being more standardized in the way they are organized, For example the Tables in Figure 1 are logically oriented with the time point in the far left and the test names over the columns and rows. This makes interpretation easy as the scales are aligned with the tests. In comparison the tables in Figure 2 have the test name on the left and the time point over the columns which is not an intuitive set-up.
- There is no comment in the analysis section of whether the data was tested distribution pattern and how that would affect the selected tests.
- Supplemental data. I am not sure all the tables or duplicate wording is necessary. For example Supp Figures 1-4 don’t really add anything that isn’t captured in some of the comparative figures later in the supplement or the main manuscript.
- I am not sure I understand how there can be significance for the pain measures in Supp Table 5 when there is absolutely no changes. It may be that in group statistics analysis this comes out, but intuitively it does not make sense. Further in the discussion this reality is identified but the comment is that the measure is “saturated” at T0. Would it m=not be more accurate to state that the measure is not sensitive in this context as other measures do detect differences.
- Conclusions. A paragraph should be added to discuss the potential utility of the two tests. While there may be a degree of concordance and alternatively a strength to using both, how will this help with prognostication – perhaps the main reason for having tests of consciousness. As acknowledged by the authors the time frame of the assessment is short, and the main emphasis was to compare the tests, but nonetheless how do they envision the tests being further validated and ultimately used.
Author Response
Please,
download the file attached.
It contains our response on a point-by-point basis.
Kind regards,
Erika Molteni
on behalf of all the authors

Reviewer 2 Report
Pediatric acquired brain trauma is a tense situation for the caregivers, clinicians and the patient. Appropriate prognosis is important as it informs decision-making (i) such as rehabilitation, (ii) family burden and expectations. This manuscript describes use of pair of neuropsychological assessments to reduce negative findings in pediatric disorders of consciousness (DoC). An important contribution to the field, the method described has greater precision and can help prevent premature uninformed detrimental decisions. The authors use well studied assessments i.e., Rappaport Coma/Near Coma Scale (CNCS) and Level of Cognitive Functioning Assessment Scale (LOCFAS) on 92 consecutive in-patients without dropouts. The authors report CNCS captured the transitions between coma levels, while LOCFAS captured the exit from coma. They observed some correlation between the assessments, and use of the combination reduced the number of false negatives from 25% to 10%. Age was not protective, surgical intervention when necessary was significant positive contributor.
Highlights
The use of consecutive patients and absence of attrition. Importance of findings to the field in general.
Limitations/revisions
It is true that the study is complex. However, writing out everything in one sentence reduces readability. There is room for improvement in writing. Writing sentences with less than 20 words could be a start. Write in third person. Several instances with “and”, “in” can be eliminated to improve readability.
- Based on the successful use of CNCS in small cohorts and validation of LOCFAS by the authors in acquired brain injury presents an opportunity to hypothesize that use of both the assessments may improve prevision.
- Measure of precision (true positives/ (true positives+false positives)) is necessary to back up the conclusion that joint assessments have greater precision.
- Present the reduction of false positives from 25% to 10% such that it is evident.
- Graphical representation of patients and their transitions within coma levels as a scatter plot will help visualize the data.
For e.g., it should be stated that the study is performed in post-acute rehabilitation period. If the data on injury-hospitalization is available, analysis if such interval relates to transition within coma levels.
Guide reader to a conclusion that is consistent with the authors’ interpretation of data, instead of stating everything in verbose sentences.
The authors have previously published in Scientific Reports and that article is much more readable than this one. Hence, use of same writing style will make this important finding accessible to wider readership.
Author Response
Dear reviewer,
please download the attached file,
containing our reply to your comments, on a point-by-point basis.
We remain available for further clarifications and changes.
Kind regards,
Erika Molteni
on behalf of all the authors
